# A Smo/Gli Multitarget Hedgehog Pathway Inhibitor Impairs Tumor Growth

**DOI:** 10.3390/cancers11101518

**Published:** 2019-10-09

**Authors:** Ludovica Lospinoso Severini, Deborah Quaglio, Irene Basili, Francesca Ghirga, Francesca Bufalieri, Miriam Caimano, Silvia Balducci, Marta Moretti, Isabella Romeo, Elena Loricchio, Marella Maroder, Bruno Botta, Mattia Mori, Paola Infante, Lucia Di Marcotullio

**Affiliations:** 1Department of Molecular Medicine, Sapienza University, Viale Regina Elena 291, 00161 Rome, Italy; ludovica.lospinososeverini@uniroma1.it (L.L.S.); irenebasili.ib@gmail.com (I.B.); francesca.bufalieri@uniroma1.it (F.B.); miriam.caimano@uniroma1.it (M.C.); elena.loricchio@uniroma1.it (E.L.); marella.maroder@uniroma1.it (M.M.); 2Department of Chemistry and Technology of Drugs, Sapienza University, Piazzale A. Moro 5, 00161 Rome, Italy; deborah.quaglio@uniroma1.it (D.Q.); silvia.balducci@uniroma1.it (S.B.); isabella.romeo1988@gmail.com (I.R.); bruno.botta@uniroma1.it (B.B.); 3CLNS@Sapienza, Istituto Italiano di Tecnologia, Viale Regina Elena 291, 00161 Rome, Italy; francesca.ghirga@iit.it; 4Department of Experimental Medicine, Sapienza University, Viale Regina Elena 324, 00161 Rome, Italy; marta.moretti@uniroma1.it; 5Department of Biotechnology, Chemistry and Pharmacy, University of Siena, via Aldo Moro 2, 53100 Siena, Italy; 6Laboratory Affiliated to Istituto Pasteur Italia-Fondazione Cenci Bolognetti—Department of Molecular Medicine, Sapienza University, Viale Regina Elena 291, 00161 Rome, Italy

**Keywords:** Hedgehog, cancer, multitarget, Smo, Gli1

## Abstract

Pharmacological Hedgehog (Hh) pathway inhibition has emerged as a valuable anticancer strategy. A number of small molecules able to block the pathway at the upstream receptor Smoothened (Smo) or the downstream effector glioma-associated oncogene 1 (Gli1) has been designed and developed. In a recent study, we exploited the high versatility of the natural isoflavone scaffold for targeting the Hh signaling pathway at multiple levels showing that the simultaneous targeting of Smo and Gli1 provided synergistic Hh pathway inhibition stronger than single administration. This approach seems to effectively overcome the drug resistance, particularly at the level of Smo. Here, we combined the pharmacophores targeting Smo and Gli1 into a single and individual isoflavone, compound **22**, which inhibits the Hh pathway at both upstream and downstream level. We demonstrate that this multitarget agent suppresses medulloblastoma growth in vitro and in vivo through antagonism of Smo and Gli1, which is a novel mechanism of action in Hh inhibition.

## 1. Introduction

Hedgehog (Hh) signaling is a developmental pathway involved in tissue homeostasis, cell stemness and tumorigenesis [1,2,3,4]. The Hh pathway activity is controlled by Hh ligands and the transmembrane receptors, Patched1 (Ptch1) and Smoothened (Smo), that are endowed with inhibitory and activator function, respectively. In the absence of Hh ligand, Ptch1 represses Smo keeping the pathway off. In the presence of ligand, Ptch1 relieves Smo-derived signals that activate Gli transcription factors, thus, promoting proliferation, survival and cell migration. Aberrant activation of the Hh pathway has been found in a wide spectrum of tumors, such as basal cell carcinoma (BCC) and medulloblastoma (MB) [5], and its deregulation can alter cancer stem cells features (i.e., self-renewal, survival, neoangiogenesis, metastatic spread) [6,7,8,9]. For this reason, Hh pathway represents an attractive target for anticancer therapy nowadays [10,11,12].

Many Hh-dependent human cancers are driven by an inappropriate upstream pathway activation (i.e., loss-of-function mutations of Ptch1 receptor, gain-of-function mutations involving Smo receptor, Hh ligand overproduction) [13,14]; that is why most of the Hh inhibitors developed to date act at an upstream level on the Smo receptor [15,16]. Unfortunately, these agents have shown many limits [17,18], such as low pharmacokinetic properties and severe side effects, including nausea, diarrhea, muscle cramping dysgeusia, fatigue, alopecia, hyponatremia, weight loss [19,20,21]. Vismodegib (Erivedge^®^, Genentech) and Sonidegib (Odomzo^®^, Novartis) are two Smo-antagonists approved by the Food and Drug Administration (FDA) for the treatment of metastatic and/or locally advanced BCC [22]. Despite some promising clinical response, treatments with Vismodegib or Sonidegib have resulted in aggressive tumor relapse, due to the development of resistant tumor clones [14,21,23,24].

Vismodegib resistance is prevalently associated with Smo mutation and to a lesser extent with alterations of downstream Hh pathway components (SuFu and Gli2) or induction of alternative signaling pathways leading to Gli1 activation, such as phosphatidylinositol 3-kinase (PI3K) and atypical protein kinase C ι/λ (aPKC-ι/λ) [24].

It is known that the Hh pathway can also be activated by Smo-independent mechanisms (i.e., gene amplification of Gli factors, mutation of the tumor suppressor SuFu, post-synthetic modifications like decreased ubiquitination-mediated degradation or acetylation of the Gli proteins or increased PI3K/mTOR/S6K1 kinase-dependent phosphorylation) [25,26,27,28,29,30,31,32,33] that lead to an increased activity of downstream Gli effectors. All mechanisms that act downstream of Smo are alternative causes of drug-resistance to the treatment of Hh-driven cancers [34,35].

One of the most relevant Hh-dependent tumors is MB, a highly aggressive and heterogeneous pediatric brain tumor with a poor clinical outcome [36]. MB has been classified in four major molecular subgroups (WNT, SHH, Group 3 and Group 4), each one presenting distinct genetic alterations, gene expression landscape and prognosis. Current treatments of MB consist of maximal surgical removal of tumor followed by radio- and chemo-therapy that have long-term toxicity effects, including delays in physical and cognitive development, a high frequency of relapse and increased cardiac risk of diseases [37,38,39].

These issues raise the need to develop new therapeutic strategies, especially focusing on the targeting of Hh-signaling. Indeed, the SHH-MB subgroup comprises 25–30% of all MB cases and is the best genetically characterized. The use of the Smo-antagonist Vismodegib to treat metastatic MB caused tumor relapse in one patient within three months, due to D473H point mutation that rendered Smo insensitive to the drug [14,19]. Although two phase II studies suggest the efficacy of Vismodegib in recurrent MB [40], the most effective strategy to overcome the drug-resistance appears the development of new Hh-inhibitors that counteract this pathway downstream of Smo or independently by Smo, as well as to combine different drugs, in a multitargeting approachable to inhibit Hh-signaling at multiple levels.

In this scenario, particular interest has emerged for the development of small molecules as inhibitors of Gli1 transcription factor, the final and most powerful positive effector of Hh signaling [41]. We have identified and characterized the isoflavone Glabrescione B (GlaB) as a potent and direct inhibitor of Gli1/DNA interaction, with strong anticancer efficacy against the Hh-dependent tumorigenesis of BCC and MB [42]. Recently, we exploited the high versatility of the isoflavone scaffold for targeting the Hh signaling pathway at multiple levels [43]. In particular, we designed and synthesized isoflavones bearing bulky chemical groups in the *para* position of ring B, with the aim to enhance the targeting at the level of the Smo receptor. Similarly, the introduction of bulky substitutions in the *meta* position of the same ring promoted the targeting of the downstream Gli effectors. Notably, the simultaneous administration of newly designed isoflavones targeting Smo and Gli1 provided synergistic Hh pathway inhibition, which might become relevant to increase the barrier to drug resistance, particularly at the level of Smo [43]. In this work, we have designed multitarget Hh pathway inhibitors through the combination of the most promising pharmacophores targeting Smo and Gli1 in a single and individual isoflavone. Organic synthesis and in vitro testing led to the identification of compound **22** as the most efficient multitarget Hh inhibitor that antagonizes both Smo and Gli1. This molecule showed strong inhibitory properties on Hh signaling as tested in functional and biological in vitro assays and in an in vivo model of Hh-dependent MB, thus, becoming the first small molecule able to target Hh signaling at multiple levels.

## 2. Results

### 2.1. Design, Synthesis and Functional Screening of Hh Inhibition by Isoflavones ***20***, ***21*** and ***22***

In a previous study, we demonstrated that the introduction of a bulky substituent in *meta* or in the *para* position of the isoflavone’s ring B enhanced the specific affinity of these compounds for Gli or Smo, respectively, and that their simultaneous administration provided synergistic Hh pathway inhibition [43]. In order to develop a multitarget Hh inhibitor, we selected the most promising GlaB-ring B derivatives [43] as specific Smo and Gli pharmacophores and combined them in a single and individual isoflavone, compound **20** (Figure 1). The ability of this newly synthesized isoflavone to inhibit Hh signaling was investigated by a luciferase reporter assay in which NIH3T3 Shh-Light II cells, stably incorporating a Gli-responsive firefly luciferase reporter (Gli-RE) and the pRL-TK Renilla as normalization control, were activated following the treatment with the synthetic Smo agonist SAG alone or in combination with compound **20**. However, **20** was inactive to suppress Hh signaling (Appendix A), probably due to the physicochemical features of the trifluoromethyl group. Based on these findings, we designed and synthetized two bioisosters featuring methyl (**21**) and chlororine (**22**) groups, respectively (Figure 1). For the synthesis of compounds **20–22** (Appendix A), we performed the deoxybenzoin approach, a mild and cost-effective method that allows the preparation of isoflavones [43]. Compounds **21** and **22** were tested for their inhibitory properties on Hh signaling by functional luciferase assay in NIH3T3 Shh-Light II cells as described above. Notably, **21** and **22** showed strong Hh pathway inhibition, with **22** being the most potent Hh inhibitor of this series with an IC_50_ of 0.79 µM (Figure 2A,B).

Afterwards, to prove the inhibitory activity of the two newly synthesized isoflavones **21** and **22** on Hh signaling at the downstream level, we verified their effects on Gli1 transcription activity in a Smo-independent condition. To this aim, we treated mouse embryonic fibroblasts (MEFs) transiently expressing ectopic Gli1 and a Gli-dependent luciferase reporter, with increasing amounts of the two compounds. Both molecules impinge Gli1 function directly, but not Gli1 exogenous protein levels, with **22** demonstrating a stronger effect (IC_50_ of 7.00 µM) (Figure 2C,D and Appendix A). These results clearly suggest that physicochemical features of substituents to isoflavone’s ring B might play a key role in binding to Smo, as well as to Gli1.

### 2.2. Inhibitory Effect of Compounds ***21*** and ***22*** on Hh-Active Cell Models

In order to demonstrate the capability of compounds **21** and **22** to target Hh signaling both at upstream and downstream level by acting on Smo and Gli respectively, we used Hh-active cell models, in which the loss of the main regulators of Hh signaling determines the constitutive activation of this pathway. At first, we treated with compounds **21** and **22** Ptch1^−/−^ mouse embryonic fibroblasts (Ptch1^−/−^ MEFs), in which the loss of repressive receptor *Ptch1* gene constitutively induces the activation of Hh signaling, thus, determining high expression levels of Hh target genes, including *Gli1*. The treatment with compound **22** at 1 and 2 μM for 48 h reduced the mRNA levels of *Gli1* in this cell model stronger than compound **21** used at the same concentrations (Figure 3A). Next, to verify the ability of the two isoflavones to inhibit Hh signaling independently of Smo, we tested their activity in mouse embryonic fibroblasts lacking the receptor Smoothened (Smo^−/−^ MEFs). Also in this cellular context, compounds **21** and **22**, as well as GANT61 and ATO (two well-known Gli1 inhibitors), but not the Smo antagonist Vismodegib, were able to reduce the mRNA expression levels of *Gli1*, with an increased effect of **22** compared to **21** (Figure 3B, Appendix A and Appendix A). To confirm the inhibitory properties of the two compounds on Hh signaling at Smo downstream level, we tested compounds **21** and **22** in SuFu^−/−^ MEFs, in which the loss of the Gli negative regulator *SuFu* leads to the constitutive activation of the pathway. The endogenous expression of *Gli1* was significantly reduced in these cells after treatment with the two compounds showing, also in this case, a stronger efficacy of compound **22** (Figure 3C). These data sustain the ability of the two isoflavones to inhibit the Hh signaling both at upstream and downstream levels and identify compound **22** as the most effective Hh-inhibitor among the ones we have tested. Moreover, compound **22** did not affect WNT or Jun/AP-1 activity supporting its specificity of action for Hh/Gli signaling (Appendix A).

### 2.3. Molecular Modeling Study on the Interaction between Compound ***22*** and Smo and Gli1

To further support the ability of compound **22** to interact with the respective targets Smo and Gli1, we performed molecular modeling simulations by using the computational protocols already described previously [42,43,44]. Results clearly show that compound **22** is able to fit the well-known antagonists’ site located within the heptahelical bundle of Smo, as well as the isoflavones’ binding site located within zinc-finger 4 and zinc-finger 5 of Gli1. In detail, docking of compound **22** to Smo by FRED docking program (OpenEye) highlighted a number of H-bond interactions between the molecule and Smo key residues, such as Asn219, Gln477, and Arg400 (Figure 4A). The aromatic core of the isoflavone’s ring B of 22 is π-π stacked to the side chain of Tyr394, while the *p*-chlorophenyl ring substituted in position -*para* of the isoflavone’s ring B is π-π stacked to His470 in a T-shaped configuration (Figure 4A). It is worth noting that this interaction is similar to the predicted binding mode of other isoflavones within the Smo antagonists’ site [43]. Molecular docking of compound **22** to Gli1 zinc finger was carried out by the GOLD docking program, showing that the compound is able to establish a H-bond interaction with Lys350, which has been already highlighted in a previous mutagenesis study [42], as well as with the well-known Thr374 from the nuclear localization signal [45] (Figure 4B). Additional H-bonds are established by compound **22** with Thr355 and the Zn-binding His351 (Figure 4B). Further, the binding mode of this molecule is highly comparable to that of other isoflavones [42,43].

To verify the direct action of compound **22** on Smo receptor we performed a Bodipy-Cyclopamine (BC) displacement assay using a fluorescent derivate of Cyclopamine that interacts with Smo at the level of its heptahelical bundle [46]. To this end, HEK293T cells were transfected with a vector expressing Smo WT or Smo D473H mutant, which confers resistance following Vismodegib treatment, and incubated with BC at increasing concentrations of compound **22**. As shown in Figure 4C, compound **22** displayed similar dose-dependent effects on both WT and D473H Smo, indicating its direct binding within the Cyclopamine site of Smo and suggesting its potential use for the treatment of Vismodegib-resistant tumors. 

To further confirm molecular modeling results on compound **22** to Gli1 zinc finger domain, we verified the effect on Gli1 mutated in Lys340 (K340A), a residue involved in DNA binding and transcriptional function of Gli1 [41,42]. As expected, different doses of compound **22** induced a moderated reduction of Gli1-dependent transcriptional activation in MEFs WT expressing ectopic Gli1 K340A and Gli1-dependent luciferase reporter compared to cells expressing Gli1 WT (Figure 4D). Of note, by ChIP assay in Gli1-overexpressing MEFs, we observed a significant reduction of the Gli1 recruitment into the *Ptch1* promoter following the treatment with compound **22** compared to control (Figure 4E). According to the high homology degree between the zinc finger domain of Gli1 and Gli2, compound **22** also inhibits Gli2-mediated transcription (Appendix A).

In summary, molecular modeling and biological assays further substantiate the multitarget effect of compound **22** by showing that the small molecule is able to fit the ligand-binding site in Smo and Gli1, and to interact with residues that are crucial for Hh signaling transduction, without affecting ciliogenesis processes or Gli1 subcellular localization (Appendix A, respectively). 

### 2.4. Compound ***22*** Inhibits the Hh-Dependent Tumor Growth In Vitro

The aberrant activation of Hh signaling is strongly involved in the onset of several tumors, such as medulloblastoma (MB). In order to test the efficacy of isoflavone **22** to suppress the Hh-dependent tumor growth, we used primary MB cells freshly isolated from Math1-cre/Ptch^C/C^ mice that spontaneously developed MB and tested in short-term cultures to keep Hh sensitivity in vitro. As shown in Figure 5A, compound **22** used at final concentrations of 0.5, 1 and 5 μM, significantly inhibits the proliferation of primary MB cells in a dose- and time-dependent manner. This was consistent with increased cell death (Figure 5B). Of note, this effect was not observed in Hh-independent MB cell lines following treatment with compound **22** (Appendix A) [47], indicating its selectivity to impair the growth of Hh-MB cells. Consistent with above data, compound **22** reduced *Gli1* and other Hh pathway target genes (Figure 5C,D) and the endogenous protein levels of Gli1 and Gli2, but not those of other Hh pathway regulators, such as ERAP1 [48], Itch [49], HDAC1 [30] and β-Catenin [50] (Figure 5E,F).

The ability of compound **22** to inhibit tumor cells proliferation and promote cell death was confirmed in Med1-MB cell line generated from a spontaneous tumor arisen in a *Ptch1*^+/−^; *lacZ mouse* [51,52] (Figure 6A,B). As expected, the treatment with compound **22** significantly reduced mRNA expression of Hh target genes and endogenous Gli1 protein levels (Figure 6C,D). To elucidate the anti-proliferative effect of isoflavone **22** also in human cancer cells we used MB Daoy cells, belonging to the SHH-MB subgroup [47,53,54]. Compound **22** showed marked activity in inhibiting cell proliferation and promoting cell death also at the lower concentration (Figure 6E,F) as a consequence of the reduction of the Hh pathway activity (Figure 6G,H).

### 2.5. Compound ***22*** Inhibits the Hh-Dependent Tumor Growth In Vivo

These promising results prompted us to test the ability of isoflavone **22** to inhibit the Hh-driven tumor growth also in an in vivo allograft model of MB. To this aim, nude mice were grafted with spontaneous primary MB cells from Math1-cre/Ptch^C/C^ mice and treated every second day with s.c. injections of compound **22** at a concentration of 5 mg/kg or solvent only. The tumor growth was monitored by caliper during the treatment period, and the tumor mass volumes were measured after explant appearing significantly reduced in compound **22**-treated mice compare to controls (Figure 7A,B). These data correlate with the inhibition of both mRNA and protein expression levels of endogenous Hh target genes in treated masses, whereas, protein levels of cleaved PARP are increased suggesting apoptotic cell death (Figure 7C–E). Moreover, compound **22**-treated tumor masses showed reduced cellularity with few MB cells dispersed in a large amount of Masson’s staining-mediated blue-labeled connective tissue (Figure 7F), associated with a decreased amount of Ki67 positive cells (Figure 7F,G). These findings support the strong anti-tumor activity of compound **22** in Hh-dependent MB and underline as the use of one molecule with multitargeting properties represents a promising therapeutic strategy to antagonize Hh-signaling and the tumor growth driven by this oncogenic pathway.

## 3. Discussion

In this study, we propose a novel mechanism of action for targeting Hh signaling pathway. Specifically, taking advantage from our previous work, we have designed an isoflavone with the aim to target simultaneously the Hh pathway at both upstream and downstream level, i.e., Smo and Gli1 respectively, and identify an innovative approach to limit tumor growth.

Abnormal Hh reactivation is a hallmark of many cancers, and several germline or somatic mutations in the Hh pathway components have been documented in BCC, MB, rhabdomyosarcoma (RMS), meningioma and many other tumors. For this reason, Hh signaling has emerged in recent years as an attractive target for anticancer therapy [55], and small molecules of both natural and synthetic origin have emerged as profitable drugs that target key components of the pathway. The majority of Hh modulators developed so far act as antagonists of the upstream Smo receptor, and two Smo antagonists (Vismodegib and Sonidegib) have been approved by the FDA for the treatment of advanced or metastatic BCC. However, the use of these drugs is strongly limited by the emergence of drug-resistance and the occurrence of aberrant Hh activation downstream of Smo [56]. Hence, the most efficacy strategy to block Hh signaling in cancer appears to be the pharmacological inhibition of the final and most powerful effector Gli1 [41]. However, only the Gli antagonist Arsenic trioxide (ATO) has entered clinical evaluation so far. For these reasons, innovative, efficacy and less toxic Hh antagonists are urgently needed as therapeutic candidates for the treatment of Hh-dependent tumors. Natural compounds represent a significant resource for the discovery and development of new Hh inhibitors, as demonstrated in the case of isoflavones, derived from plants of the *Leguminosae* family [57]. In a recent study, we have exploited the high versatility of the isoflavone scaffold, and we demonstrated that the introduction of bulky chemical groups in the *para* position of the isoflavone’s ring B enhances the targeting at the level of the Smo receptor. In contrast, bulky substitutions introduced in the *meta* position of the same ring promote the targeting of the downstream Gli effectors. Simultaneous administration of isoflavones targeting Smo and Gli provided synergistic Hh pathway inhibition with a reduction of around 20 folds of the administered dose, which might be relevant to limit toxic side effects and overcome the Smo-drug resistance [43].

Here, by combining the most profitable pharmacophores for targeting Smo and Gli1 by synthetic isoflavones, we have designed and synthesized the isoflavone **22** acting as a multitarget Hh inhibitor that targets both Smo and Gli1 at the same time. Compound **22** was able to inhibit Hh-dependent tumor growth in human and murine MB cells at sub-micromolar concentration, as a consequence of the reduction in *Gli1* expression levels. Despite the limitations of intratumoral administration (i.e., the invasive nature of the injection itself, the rapid clearance of drugs directly applied to the tumor and the development of dose-limiting toxicities in the area surrounding the site of injection) isoflavone **22** remarkably showed a strong anti-tumor effect also in vivo by suppressing cell proliferation and promoting apoptosis. Molecular modeling further corroborated the multitarget mechanism of action of **22**, showing that the molecule is able to fit the ligand-binding site in both Smo and Gli1. 

Overall, these results reveal a valuable form of targeted therapy to increase efficacy and to decrease the toxicity of individual anticancer agents. Our findings discover the first multitarget Hh inhibitor that impinges the Hh-dependent tumor growth and stands as new potential weapons against Hh-driven cancer, such as medulloblastoma, the most malignant childhood brain tumor for which poor therapeutic options exist.

## 4. Material and Methods

### 4.1. General Experimental Methods for the Preparation of Compounds ***20***–***22***

All reagents were commercial and were used without further purification. Chromatography was carried out on silica gel (230–400 mesh). All reactions were monitored by thin-layer chromatography (TLC), and silica gel plates with fluorescence F254 were used. Melting points were taken in open capillaries on a Büchi Melting Point B-545 apparatus and are presented uncorrected. ^1^H NMR and ^13^C NMR spectra were recorded using a Bruker 400 Ultra ShieldTM spectrometer (operating at 400 MHz for ^1^H and 100 MHz for ^13^C) using tetramethylsilane (TMS) as an internal standard. Chemical shifts (d) are reported in parts per million (ppm) and are referenced to CHCl_3_ (7.26 ppm for ^1^H, 77.16 ppm for ^13^C). All ^13^CNMR spectra were measured with complete proton decoupling. Data for NMR spectra are reported, as follows—s (singlet), d (doublet), t (triplet), q (quartet), m (multiplet), br (broad) signal, J coupling constant in Hz. Electron spray ionization mass spectra (ESI-MS) were recorded on Bruker BioApex Fourier transform ion cyclotron resonance (FT-ICR) mass spectrometer. HPLC analysis was performed on a Waters 2690 Separation Module, equipped with a Rheodyne Model 8125 20 mL injector and a Model M486 programmable multi-wavelength detector (PDA). The purity of the sample used in this study was higher than 95% by HPLC. HPLC conditions, % and Retention times (Rt) were as follows. Column: Phenomenex Luna C18, 5.0 μm (250 × 4.6 mm). Eluent A) water/acetonitrile = 95:5 (*v/v*). Eluent B) water/acetonitrile = 5:95 (*v/v*). Gradient elution: For 0–5 min A: B = 50:50; 5–20 min up to 100% B; 20–25 min to 100% B. Flow rate: 1.0 mL/min. PDA detection at 200–400 nm. 

### 4.2. General Procedure for the Synthesis of Compounds ***20***–***22***

The procedure for the synthesis of isoflavone 4d has been already described in Berardozzi et al. [43]. To a solution of the isoflavone (4d) (0.18 mmol, 1.00 equiv.) in acetone (5 mL), K_2_CO_3_ (1.8 mmol, 10.00 equiv.) was added. After stirring for 15 min the corresponding benzyl bromide (0.9 mmol, 5.00 equiv.) was added drop wise to the mixture and stirred at 45 °C overnight. After removing the acetone in a vacuum, H_2_O (10 mL) and EtOAc (20 mL) were added, and the aqueous phase was extracted with EtOAc (3 × 20 mL). The combined organic layers were dried over Na_2_SO_4_, and finally concentrated under reduced pressure. The residue was purified by Flash Chromatography using Petroleum Ether/ EtOAc as eluent to give the corresponding substituted-isoflavone (**20**-**22**). The characterization for all compounds is reported in the SI. 

### 4.3. Luciferase Reporter Assay

The Hh-dependent luciferase assay was performed in NIH3T3 Shh-Light II cells, stably expressing a Gli-responsive luciferase reporter and the pRL-TK Renilla (normalization control), treated for 48 h with SAG (200 nM) and the tested compounds at the indicated concentrations. Luciferase and Renilla activities were assayed with a dual-luciferase assay system according to the manufacturer’s instructions (Biotium Inc., Hayward, CA, USA). Results were expressed as Luciferase/Renilla ratios and represented the mean ± S.D. of three experiments, each performed in triplicate.

### 4.4. Cell Cultures, Transfection and Treatments

NIH3T3 Shh-Light II, MEFs WT, Ptch1^−/−^ (kindly provided by M.P. Scott), SuFu^−/−^ and Smo^−/−^ (kindly provided by R. Toftgard) MEFs and Med1-MB cells (kindly provided by Yoon-Jae Cho), D425 and HDMB03 (kindly provided by V. D’Angiolella) were cultured in DMEM plus 10% FBS. Daoy cells (obtained from the American Type Culture Condition, ATCC) were cultured in Eagle’s minimum essential medium (MEM) plus 10% FBS. D458, D283 (kindly provided by V. D’Angiolella) and D341 (obtained from the American Type Culture Condition, ATCC) were maintained in Eagle’s minimum essential medium (MEM) plus 20% FBS. All media contained 1% Penicillin-Streptomycin and 1% Glutamine. Primary MB cells were freshly isolated from Math1-cre/Ptch^C/C^ mice. Tumors were collected and mechanically disrupted with fire-polished Pasteur pipettes in HBSS with 1% Pen/Strep and treated with DNase (10 μg/mL) for twenty minutes to obtain a single-cell suspension. Cells were centrifuged and resuspended in Neurobasal Media-A with B27 supplement minus vitamin A, penicillin–streptomycin (1%) and L-glutamine (1%) and used for short-term to keep Hh-sensitivity in vitro. Mycoplasma contamination in cell cultures was routinely detected by using PCR detection kit (Applied Biological Materials, Richmond, BC, Canada). Transient transfections were performed using DreamFect^TM^ Gold transfection reagent (Oz Biosciences SAS, Marseille, France). NIH3T3 Shh-Light II cells were treated with SAG (200 nM, Alexis Biochemicals Farmingdale, NY, USA) for 48 h. Where indicated, cells were treated with Vismodegib (Selleckchem), Arsenic (III) oxide (ATO, Sigma Aldrich, St. Louis, MO, USA) and GANT61 (Enzo Life Sciences, Exeter, UK) at the indicated concentrations. 

### 4.5. Bodipy-Cyclopamine (BC) Binding Assay

Human Myc-DDK-tagged Smo WT or human Myc-DDK-tagged Smo D473H was transfected in HEK293T cells. Cells were washed in PBS supplemented with 0.5% fetal bovine serum, fixed in 4% paraformaldehyde in phosphate-buffered saline (PBS) for 10 min, and incubated for 2 h at 37 °C both in the same medium supplemented with Bodipy-Cyclopamine (5 nM) and the studied compounds at indicated concentrations. The cells were permeabilized with 0.2% Triton X100 (Sigma) 0.2%. Dako Fluorescent mounting (Dako, Carpinteria, CA, USA) was used as a mounting medium and Hoechst reagent for staining of the cell nuclei. Bodipy (green) and Hoechst (blue) signals were analyzed.

### 4.6. mRNA Expression Analysis

Trizol reagent (Invitrogen/Life Technologies, Carlsbad, CA, USA) was used to isolate total RNA from cells and tissues and reverse transcribed with SensiFAST cDNA Synthesis Kit (Bioline Reagents Limited, London, UK). Quantitative real-time PCR (Q-PCR) analysis of *Gli1*, *Gli2*, *Ptch1*, *Hip1*, *N-Myc*, *PCNA*, *β-2 microglobulin* and *Hprt* mRNA expression was performed on each cDNA sample using the VIIA7 Real Time PCR System employing Assay-on-Demand Reagents (Life Technologies). FAST Q-PCR thermal cycler parameters were used to amplify the reaction mixtures containing cDNA template, SensiFAST Probe Lo-ROX Kit (Bioline Reagents Limited, London, UK) and primer probe. Each amplification reaction was performed in triplicate, and the average of the three threshold cycles was used to calculate the amount of transcript in the sample (using SDS version 2.3 software). mRNA quantification was expressed as the ratio of the genes of interest quantity to the housekeeping genes quantity. All values were normalized with two endogenous controls, *β-2 microglobulin* and *Hprt*.

### 4.7. Cell Proliferation Assay

To determine the growth rate of viable MB cells, a trypan blue count was performed after a treatment period of 24–48–72 h with the studied compounds or solvent only used as control.

### 4.8. Immunoblot Analysis

Tissues were lysed in a solution containing RIPA buffer (50 mM Tris-HCl at pH 7.6, 150 mM NaCl, 0.5% sodium deoxycholic, 5 mM EDTA, 0.1% SDS, 100 mM NaF, 2 mM NaPPi, 1% NP-40) supplemented with protease and phosphatase inhibitors. Lysates were centrifuged at 13,000 × *g* for 30 min at 4 °C and the resulting supernatants were subjected to immunoblot analysis with the following antibodies: mouse anti-Gli1 (L42B10, 1:500) and rabbit anti-PARP (9542, 1:1000) purchased from Cell Signaling (Beverly, MA, USA); rabbit anti-Cyclin D1-20 (sc-717, 1:500), rabbit anti-β-Catenin (sc-7199, 1:1000) and goat anti-Actin (sc-1616, 1:1000) were purchased from Santa Cruz Biotechnology (Santa Cruz, CA, USA); mouse anti-ERAP1 6H9 (1:1000) kindly provided by P. van Endert; goat anti-Gli2 (AF3635, 1:1000) was purchased from R&D Systems; mouse anti-Itch (611199, 1:1000) antibody was purchased from BD Bioscience (Heidelberg, Germany); rabbit anti-HDAC1 (H3284, 1:1000) was purchased from Sigma Aldrich (St. Louis, MO, USA).

### 4.9. Chromatin Immunoprecipitation

MEFs WT transfected with Flag-tagged Gli1 plasmid or pcDNA3.1 were crosslinked, and chromatin immunoprecipitation was carried out with (1:200) mouse anti Flag-M2 antibody (Sigma Aldrich, St. Louis, MO, USA). Eluted DNA was analyzed by qRT-PCR as previously described [42].

### 4.10. Subcellular Fractionation

Freshly harvested cells were lysed in Buffer A (10 mM HEPES at pH 7.4, 10 mM KCl, 10 mM NaCl, 0.1 mM EDTA, 0.1 mM EGTA, 1 mM DTT, 0.5 mM PMSF) and centrifuged at 11,000 × g for 20 min to obtain the cytoplasmic fraction. The nuclear pellet was washed in Buffer B (20 mM HEPES at pH 7.4, 20% glycerol, 100 mM KCl, 1 mM EDTA, 1 mM DTT, 0.5 mM PMSF, 10 ng ml^−1^ Leupeptin) and centrifuged at 11,000 × g for 10 s to remove the supernatant. Nuclei were extracted with Buffer C (20 mM HEPES at pH 7.4, 20% glycerol, 400 mM NaCl, 1 mM EDTA, 1 mM EGTA, 1 mM DTT, 0.5 mM PMSF, 10 ng ml^−1^Leupeptin) by centrifugation at 13,000 × g for 10 min. Lysates were analyzed by western blotting assay.

### 4.11. Immunofluorescence

For the analysis of ciliogenesis, MEFs WT were treated with compound **22** at several concentrations for 24 h in 0.05% FBS and then fixed 10 min in 4% paraformaldehyde. The primary antibody for the cilium marker IFT88 (Rabbit Polyclonal 13967-1-AP, from ProteinTech, Manchester, UK) was incubated overnight at the dilution (1:200) in blocking solution (2% FBS, 2% BSA, 0.2% fish gelatin in PBS1X). A secondary antibody conjugated to Alexa-488 (ThermoFisher Scientific, MA, USA) and DAPI for the staining of nuclei, were diluted in blocking solution (1:1000) and incubated 1 h at room temperature. Images were acquired through 60X oil objective lens (Olympus IX81) coupled to a monochrome CCD camera (Sensicam QE; Cooke Corporation, ME, USA). For quantification of IFT88 positive cells and the length of cilia expressed in µm was used ImageJ Measure function to count the number of objects and their length.

### 4.12. Immunohistochemistry

For immunohistochemical staining tissues were fixed in formalin and paraffin embedded. Sections were incubated with rabbit monoclonal Ki67 antibody (Thermo Fisher Scientific, MA, USA) (1:100) diluted in PBS. Detection was carried out with the mouse-to-mouse HRP (DAB) staining system (ScyTek Laboratories, Logan, UT, USA) accordingly to the manufacturer’s instructions.

### 4.13. Molecular Modeling

The predicted binding mode of compound **22** to Smo and Gli1 was investigated as described previously [43]. Briefly, molecular docking to Smo was carried out with FRED version 3.3.0.3 (OpenEye) (OpenEye Scientific Software, Santa Fe, NM, USA. http://www.eyesopen.com) [58,59], using the highest docking resolution, while other parameters were used at their default values. Ligand conformational analysis was carried out with OMEGA version 3.1.0.3 (OpenEye) (OpenEye Scientific Software, Santa Fe, NM, USA. http://www.eyesopen.com) [46] by storing up to 600 conformations of the molecule. The crystallographic structure of Smo in complex with cyclopamine coded by PDB-ID 4O9R was used as a rigid receptor in docking towards Smo [46]. Docking to Gli1 zinc finger domain was carried out by GOLD version 5.7.1 (The Cambridge Crystallographic Data Centre, Cambridge, UK) [60] using the Goldscore docking function. The crystallographic structure of Gli1 in complex with DNA coded by PDB-ID 2GLI was used as a rigid receptor in docking towards Gli1 zinc finger domain [61].

### 4.14. Animal Studies

For allograft experiment, spontaneous MB from Math1-cre/Ptch^C/C^ mice was collected, minced and pipetted to obtain a single-cell suspension. Equal amounts of cells (2 × 10^6^) were injected s.c. at the posterior flank of BALB/c nude mice (*nu/nu*) (Charles River Laboratories, Lecco, Italy). When tumors reached a median size of ~150 mm^3^, animals were randomly divided into two groups (*n* = 6) and intratumorally injected every second day with compound **22** or solvent only (2-hydroxypropyl-b-cyclodextrin:DMSO) for 18 days. Tumor growth was monitored and measured with caliper. Changes in tumor volume were evaluated with the formula (length × width) × 0.5 × (length + width). All animal protocols were approved by local ethic authority (Ministry of Health) and conducted in accordance with Italian Governing Law (D.lgs 26/2014).

### 4.15. Statistical Analysis

Statistical analysis was performed using the StatView 4.1 software (Abacus Concepts, Berkeley, CA, USA). For in vivo studies, statistical differences were analyzed by Mann-Whitney *U*-test for non-parametric values, and a *p* value < 0.05 was considered significant. For all other experiments, *p* values were determined using two-tailed Student’s *t-*test, and statistical significance was set at *p* < 0.05. Results are expressed as mean ± S.D. from an appropriate number of experiments (at least three biological replicas).

## 5. Conclusions

We designed a modified isoflavone bearing specific substitutions at *para* or *meta* position of ring B that are preferred for the interaction with Smo or Gli1, respectively. We demonstrated that this small molecule, compound **22**, is able to target the Hh pathway at both upstream and downstream level simultaneously, leading to a marked tumor growth inhibition in a model of Hh-dependent cancer. Our study provides significant support in oncology research for the development of new clinically relevant Hh inhibitors, and encourages the use of a multitargeting approach for the treatment of Hh-driven tumors.

## Figures and Tables

**Figure 1 cancers-11-01518-f001:**
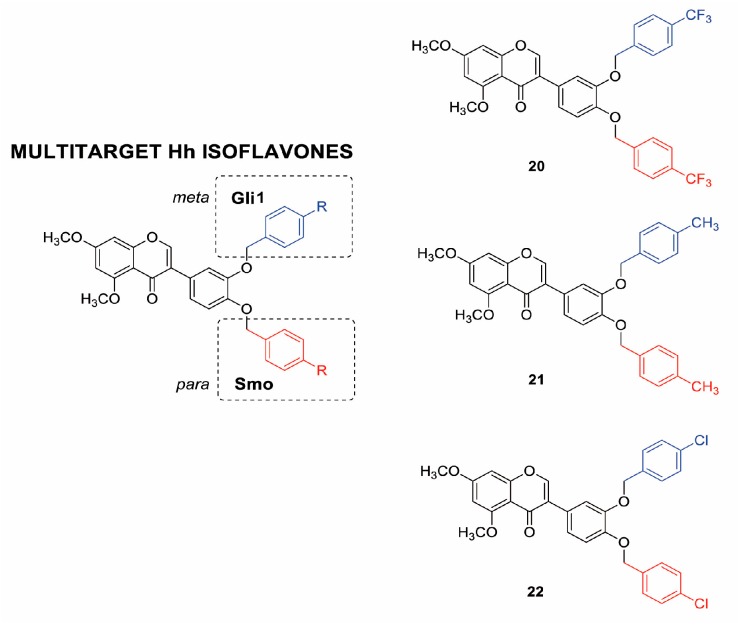
Chemical structure of isoflavones **20–22**. GlaB-ring B derivatives were designed as multitarget Hh inhibitors and synthesized via deoxybenzoin route. O-substitution at *meta* position of ring B (blue) is preferred to interact with Gli, whereas, O-substitution at *para* position (red) is preferred for the interaction with Smo.

**Figure 2 cancers-11-01518-f002:**
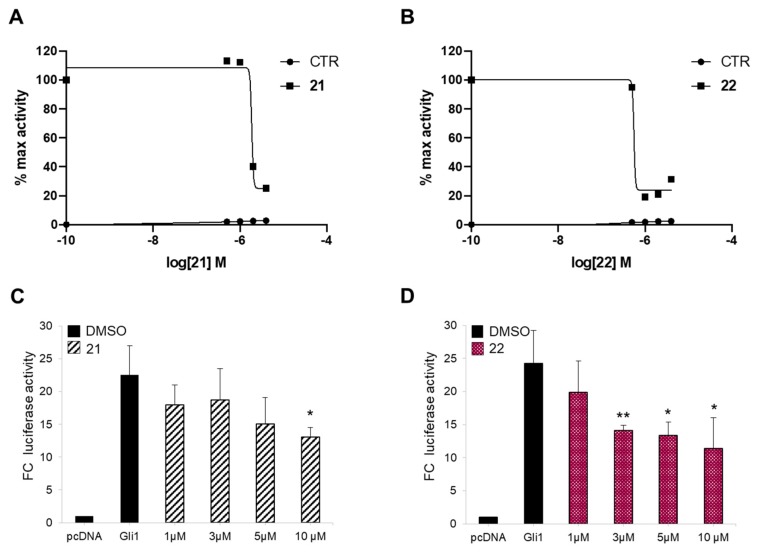
Hh inhibition by compounds **21** and **22**. The dose-response curve in SAG-treated NIH3T3 Shh-Light II cells (**A,B**) or mouse embryonic fibroblasts (MEFs) transfected with 12XGliBS-Luc and pRL-TK Renilla (normalization control) plus control (empty) or Gli1 vector (**C,D**). Cells were treated with increasing concentrations of compounds **21** (**A,C**) and **22** (**B,D**). Treatment time was 48 h and 24 h for NIH3T3 Shh-Light II cells and transfected MEFs, respectively. Data were normalized against Renilla luciferase. Data show the mean ± SD of three independent experiments. (*) *p* < 0.05 vs. SAG or Dimethyl sulfoxide (DMSO); (**) *p* < 0.01 vs. SAG or DMSO.

**Figure 3 cancers-11-01518-f003:**
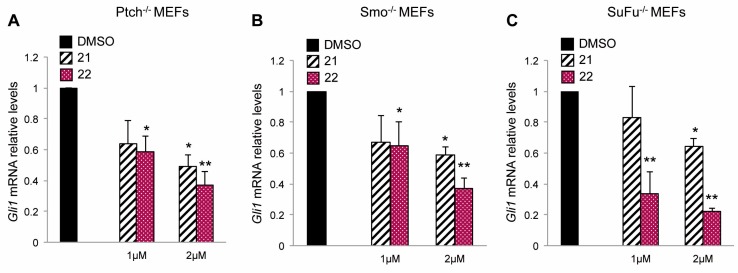
Inhibitory effect of compounds **21** and **22** on Hh-active cell models. The graphs show mRNA expression levels of *Gli1* in Ptch1^−/−^ (**A**), Smo ^−/−^ (**B**) and SuFu^−/−^ (**C**) MEFs treated for 48 h with DMSO as a control, or compounds **21** or **22** at 1 or 2 μM concentrations. *Gli1* mRNA levels were normalized to endogenous controls *β2-microglobulin* and *Hprt*. Data show the mean ± SD of three independent experiments. (*) *p* < 0.05 vs. DMSO; (**) *p* < 0.01 vs. DMSO.

**Figure 4 cancers-11-01518-f004:**
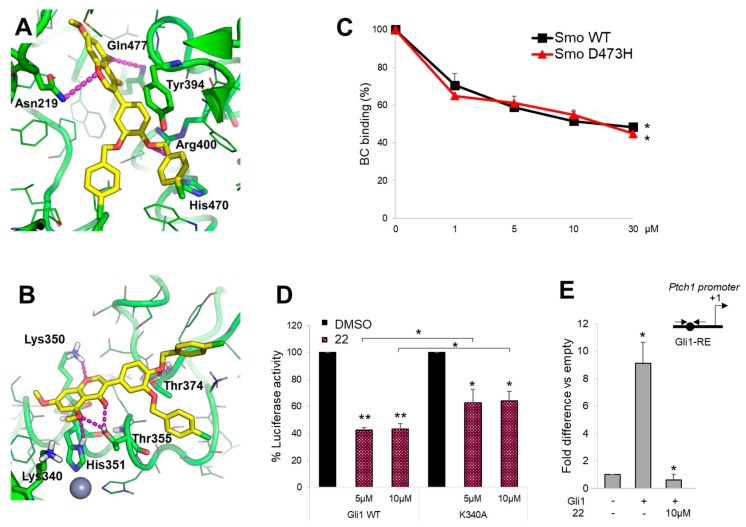
Compound **22** directly binds Smo and Gli1. (**A**,**B**) Predicted binding mode of compound **22** to Smo and Gli1. The small molecule is shown as yellow sticks, the proteins as a green cartoon; residues within 5 Å from the ligand are shown as lines. Residues contacted by compound **22** are shown as sticks and are labeled. Lys340 is shown as sticks as well. H-bond interactions are highlighted by magenta dashed lines. (**C**) The concentration-response curves show the percentage of BC binding to WT or D473H Smo mutant after compound **22** treatment. (*) *p* < 0.05 vs. CTR. (**D**) MEFs WT were transfected with 12XGliBS-Luc and pRL-TK Renilla (as normalization control) plus empty vector or WT or K340A Gli1 mutant; 24 h after transfection cells were treated with DMSO only or increasing concentrations of compound **22**. Luciferase activity was analyzed 24 h after treatment. (*) *p* < 0.05 vs. DMSO, (**) *p* < 0.01 vs. DMSO; (*) *p* < 0.05 K340A vs. WT Gli1. (**E**) MEFs WT were transfected with Flag-tagged Gli1 or empty vector and then treated with compound **22** at 10 μM. Chromatin immunoprecipitation (ChIP) and qRT–PCR using primers surrounding the Gli1-binding site (BS) of mouse *Ptch1* promoter (schematic representation on the right) were performed. Results are indicated as fold difference to empty control. All data show the mean ± S.D. of three independent experiments. (*) *p* < 0.05 vs. empty control.

**Figure 5 cancers-11-01518-f005:**
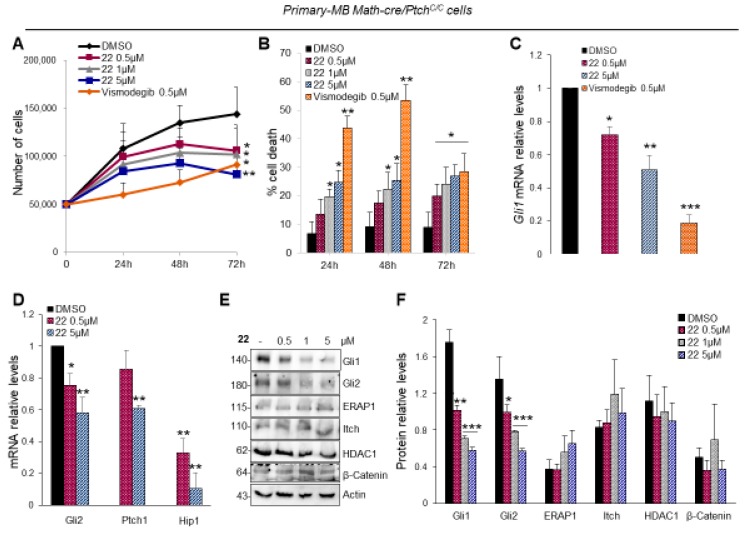
Hh-dependent tumor growth inhibition in vitro by compound **22**. (**A,B**) Primary cell cultures from Math1-cre/Ptch^C/C^ mice medulloblastoma (MBs) were treated with compound **22** (0.5, 1 and 5 μM), Vismodegib (0.5 μM) or DMSO only. After the indicated times, a trypan blue count was performed to determine the growth rate of viable cells and the percentage of cell death. (**C,D**) *Gli1* and Hh target genes mRNA levels of primary MB cells treated with compound **22** determined by qRT–PCR and normalized to endogenous control *β2-microglobulin* and *Hprt*. (**E**,**F**) Representative immunoblotting and densitometric analyses of the indicated proteins in primary MB cells after the treatment with compound **22**. Data show the mean ± SD of three independent experiments. (*) *p* < 0.05 vs. DMSO; (**) *p* < 0.01 vs. DMSO; (***) *p* < 0.001 vs. DMSO.

**Figure 6 cancers-11-01518-f006:**
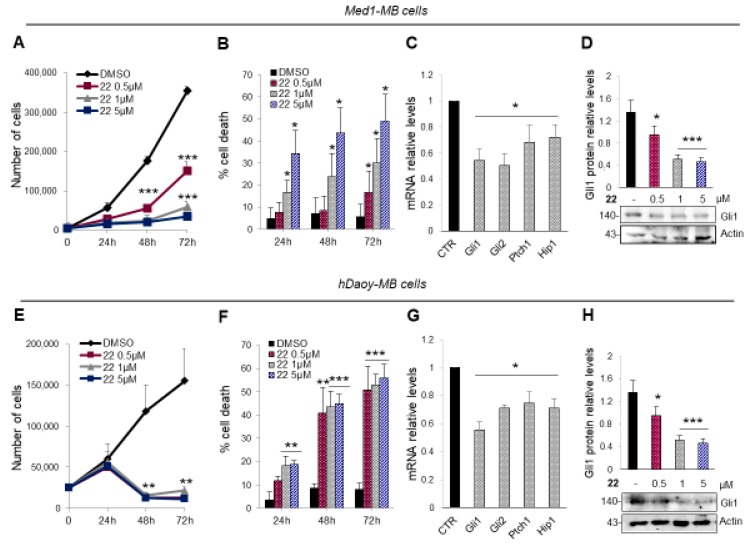
Mouse Med1-MB (**A**–**D**) and human Daoy MB (**E**–**H**). Cells were treated with compound **22** (0.5, 1 and 5 μM) or DMSO only. After the indicated times, a trypan blue count was performed to determine the growth rate of viable cells and the percentage of cell death (**A**,**B** and **E**,**F**). mRNA expression levels of the Hh target genes (**C**,**G**) and Gli1 protein levels with the corresponding densitometric analysis (**D**,**H**) were shown. Data show the mean ± SD of three independent experiments. (*) *p* < 0.05 vs. DMSO; (**) *p* < 0.01 vs. DMSO; (***) *p* < 0.001 vs. DMSO.

**Figure 7 cancers-11-01518-f007:**
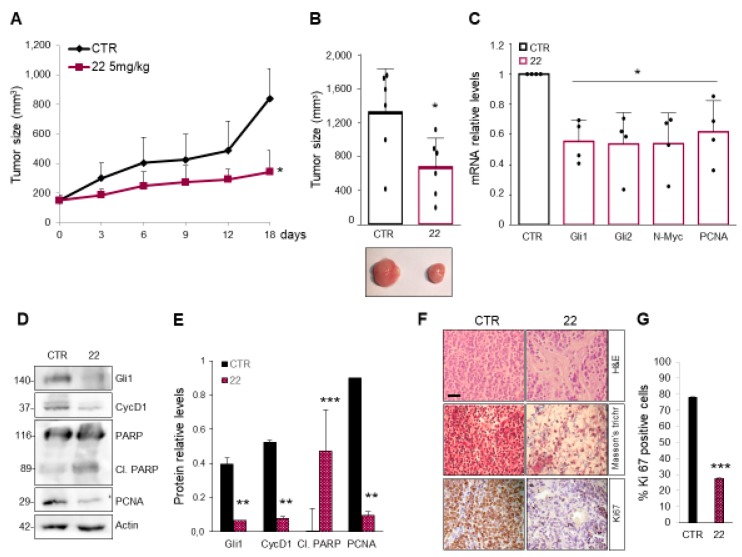
Compound **22** impairs Hh-dependent tumor growth in vivo. (**A**) BALB/c nude mice (*nu/nu*) were grafted with spontaneous primary MB from Math1-cre/Ptc^C/C^ mice. Tumor masses (150 mm^3^) were injected with compound **22** or solvent only. Tumor growth was monitored during the treatment period. (**B**) Representative flank allograft average volumes (lower panel) and quantification of tumor explants (upper panel). (**C**) The graphs show mRNA expression levels of *Gli1*, *Gli2*, *N-Myc* and *PCNA* determined by qRT–PCR and normalized to endogenous control *β2-microglobulin* and *Hprt*. (**D**,**E**) Western blot and densitometric analysis show Hh target protein and proliferation marker expression levels from the tumor masses assayed in **B**. (**F**) Hematoxylin/Eosin (H&E), Masson’s trichrome and Ki67 staining of representative tumor masses. Scale bar 100 μm. (**G**) Quantification of Ki67 immunohistochemical staining, shown in **F**. Data show the mean ± S.D. of tumor (*n* = 6) for each treatment. (*) *p* < 0.05 vs. CTR; (**) *p* < 0.01 vs. CTR; (***) *p* < 0.001 vs. CTR.

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
