# Peer review of "A Smo/Gli Multitarget Hedgehog Pathway Inhibitor Impairs Tumor Growth"

_cancers, 2019, doi:10.3390/cancers11101518_

Round 1

Reviewer 1 Report

The study of Severini et al. describes the synthesis and characterisation of an isoflavone-based dual SMO/GLI1 inhibitor. The works stems from a published report showing that para or meta  substitutions in the isoflavone B ring confer selectivity towards SMO or GLI1. This study investigates the effects of a dually-modified isoflavone in canonical Hh signalling in cultured cells and in preclinical medulloblastoma models.

While the findings are of therapeutic interest, a more detailed characterisation of the mechanism of action of the isoflavones is required for publication. The manuscript would be improved by the following:

1- a brief description of MB subtypes and the % of those that belong to the Shh-subtype should be mentioned in the introduction. The way it is written seems to suggest that all MB patients treated with vismodegib developed resistance through a unique mutation D473H, while this finding was reported in only 1 patient. Moreover, as there are publications of clinical trials suggesting efficacy of vismodegib in some MB contexts.

2- Dose response curves to an inhibitory compound are usually shown as % maximal activity vs. log [M] instead of a bar graph format. This is useful to infer form the fitting curve the type of pharmacological mechanism.

3- Compound 21 and 22 have a modest effect on Gli-luciferase activity of MEFs transfected with GLI1: is the expression level and stability of GLI1 affected by the compound? At least a western blot analysis of GLI1 expression should be provided. 

4- In addition, Fig. 2A and B should include a vehicle-stimulated data point, and Fig 2C and D could include the negative control values (empty vector) for the reader to interpret the extent of stimulation with SAG or GLI1 and or inhibition by the compounds 21 and 22. 

5- SMO-/- MEFs express almost undetectable levels of GLI1 mRNA, unlike PTCH1-/- MEFs. Can the authors list in supplementary information the ΔCt values for basal levels of Gli1 in DMSO-treated and isoflavone treated conditions?

6- I cannot comment on the molecular modelling data as this is not my area of expertise. However, can the authors explain with agonist-binding site of MSO they refer to? SMO is known to bind oxysterol cholesterol and alkaloids in different positions. Similarly, can the authors describe in which TM domain or EC/IC loop are Asn 219, Gln477 and Arg400 located?

7- The experiments in MB primary and established cell lines should include GLI1 mRNA and protein expression, as well as a second GLI-dependent target gene such as HHIP, for all the cell models. The cytotoxic effect of compound 22 appears to be due to additional targets other than the Hh pathway since the reduction of GLI1 is at best 50% in primary MB cells. Importantly, the use a well-characterised SMO antagonist like vismodegib or cyclopamine could shed light on potential off target effects. Alternatively, unbiased screens like by RNA-seq will inform the audience of the pathways affected by compound 22.

8- In the xenograft experiments, Fig 6B and C would be better presented by showing each individual data point in addition to the mean and S.D.

9- the study will be further improved by investigating the effect of compound 22 administration in the prevention of MB formation in Math1-cre/Ptc1C/C mice. The experiments seems feasible since by 10 weeks of age, all animals develop MB (Z-J Yang et al. Cancer Cell 2008).

Minor corrections:

1- please revise the manuscript for use of English. For example, "This approach stands as a relevant to overcome the drug resistance... " for "This approach seems to effectively overcome drug resistance..."; "in a single and individual" for "into a single"; "antagonism at both Smo and Gli1" for "antagonism of Smo and Gli1";  "The Hedgehog signalling...." for "Hedgehog signalling or The Hedgehog pathway"; "Hh pathway" for "the Hh pathway". There are many more in accuracies that need to be fixed.

2- Please add a paragraph in the introduction  describing the canonical Hh pathway for the reader's benefit.

3- Vismodegib resistance by activation of PKCiota should be included in the introduction.

Author Response

Reviewer #1

Open Review

English language and style

( ) Extensive editing of English language and style required
(x) Moderate English changes required
( ) English language and style are fine/minor spell check required
( ) I don't feel qualified to judge about the English language and style

Yes

Can be improved

Must be improved

Not applicable

Does the introduction provide sufficient background and include all relevant references?

( )

( )

(x)

( )

Is the research design appropriate?

( )

(x)

( )

( )

Are the methods adequately described?

(x)

( )

( )

( )

Are the results clearly presented?

( )

(x)

( )

( )

Are the conclusions supported by the results?

( )

( )

(x)

( )

Comments and Suggestions for Authors

The study of Severini et al. describes the synthesis and characterisation of an isoflavone-based dual SMO/GLI1 inhibitor. The works stems from a published report showing that para or meta  substitutions in the isoflavone B ring confer selectivity towards SMO or GLI1. This study investigates the effects of a dually-modified isoflavone in canonical Hh signalling in cultured cells and in preclinical medulloblastoma models.

While the findings are of therapeutic interest, a more detailed characterisation of the mechanism of action of the isoflavones is required for publication.

We thank the Reviewer for the positive general comments on our study.

The manuscript would be improved by the following:

1- a brief description of MB subtypes and the % of those that belong to the Shh-subtype should be mentioned in the introduction. The way it is written seems to suggest that all MB patients treated with vismodegib developed resistance through a unique mutation D473H, while this finding was reported in only 1 patient. Moreover, as there are publications of clinical trials suggesting efficacy of vismodegib in some MB contexts.

As requested by the Reviewer, these points have been included in the revised version of the manuscript.

2- Dose response curves to an inhibitory compound are usually shown as % maximal activity vs. log [M] instead of a bar graph format. This is useful to infer form the fitting curve the type of pharmacological mechanism.

As requested by the Reviewer, dose response curves have been modified in the revised version of the manuscript (revised Figure 2A,B).

3- Compound 21 and 22 have a modest effect on Gli-luciferase activity of MEFs transfected with GLI1: is the expression level and stability of GLI1 affected by the compound? At least a western blot analysis of GLI1 expression should be provided. 

As requested by the Reviewer, a western blot analysis of Gli1 has been performed in MEFs overexpressing Gli1 after treatment with compounds (new Supplementary Figure S3) and included in the revised manuscript. Data show that compounds 21 and 22 do not affect exogenous Gli1 protein levels. On the other hands, compound 22 significantly reduces endogenous expression of Gli1 in a dose-response manner in primary MB Ptch-/-, Med1 and Daoy cells (revised Figures 5E, new Figures 5D and 5H).

4- In addition, Fig. 2A and B should include a vehicle-stimulated data point, and Fig 2C and D could include the negative control values (empty vector) for the reader to interpret the extent of stimulation with SAG or GLI1 and or inhibition by the compounds 21 and 22. 

As requested by the Reviewer, these figures have been modified in the revised version of the manuscript (revised Figure 2A-D).

5- SMO-/- MEFs express almost undetectable levels of GLI1 mRNA, unlike PTCH1-/- MEFs. Can the authors list in supplementary information the ΔCt values for basal levels of Gli1 in DMSO-treated and isoflavone treated conditions?

We agree with the Reviewer that the Gli1 mRNA in Smo-/- MEFs is lower than in Ptch-/- MEFs, however its levels are detectable and can be modulated (new Supplementary Table 1). In this regard, we observed that our compounds, as well as GANT61 and ATO (two well known Gli inhibitors), but not Vismodegib, reduce Gli1 mRNA levels in Smo-/- MEFs, thus proving the inhibitory activity of isoflavones 21 and 22 on Hh signaling independently of Smo (new Supplementary Table 1 and Figure S4)

6- I cannot comment on the molecular modelling data as this is not my area of expertise. However, can the authors explain with agonist-binding site of MSO they refer to? SMO is known to bind oxysterol cholesterol and alkaloids in different positions. Similarly, can the authors describe in which TM domain or EC/IC loop are Asn 219, Gln477 and Arg400 located?

We agree with the Reviewer that SMO has two main binding site for small molecules, characterized by X-ray crystallography, which are located in the extracellular cysteine-rich domain (CRD) or in the transmembrane domain (TM) in the so named heptahelical bundle. The first site is known to bind oxysterols, although a recent report shows the binding of an oxysterol within the TM site by Cryo-EM methods (see Nature volume 571, pages279–283 (2019)). The second site is located within the extracellular portion of the TM domain of SMO, and it is formed by the mutual arrangement of transmembrane alpha helices such as described in a number of structural biology reports (i.e. (2013) Nature 497 338-343; (2014) Nat Commun 5 4355-4355; (2016) Nature 535 517-522).

7- The experiments in MB primary and established cell lines should include GLI1 mRNA and protein expression, as well as a second GLI-dependent target gene such as HHIP, for all the cell models. The cytotoxic effect of compound 22 appears to be due to additional targets other than the Hh pathway since the reduction of GLI1 is at best 50% in primary MB cells. Importantly, the use a well-characterised SMO antagonist like vismodegib or cyclopamine could shed light on potential off target effects. Alternatively, unbiased screens like by RNA-seq will inform the audience of the pathways affected by compound 22.

As requested by the Reviewer, protein levels of Gli1 and expression analysis of Gli1 and Gli-dependent genes, such as Ptch and Hhip, in primary MB cells (revised Figure 5C, new Figure 5D and 5E) and in other Hh-dependent cell lines (Med1 and Daoy) (new Figure 6) have been included in the revised version of the manuscript. Compound 22 reduces Hh-targets genes and Gli1 protein levels in all tested cell models (revised Figure 5C, new Figure 5D, 5E and new Figure 6C,D and G,H). Further, primary MB cells treated with compound 22 showed a significant inhibition of the proliferation in comparison with the well known Smo antagonist Vismodegib (revised Figure 5). Although we cannot exclude the activity of isoflavone 22 on additional target, this compound does not affect luciferase activity driven by Hh-unrelated (i.e. Jun/AP1) and Hh-related (i.e. Wnt/β-catenin) pathway (new Supplementary Figure 5), and does not affect protein levels of others Hh regulators (i.e. HDAC1, ERAP1, Itch, β-catenin) (new Figure 5E), further indicating its selectivity for Hh/Gli signaling.

RNASeq, as suggested by the reviewer, could definitely give many information about the possible effect of compound 22 on other signaling pathways. Nevertheless, this would require many replicates, especially for tumor primary cell culture, to strengthen the statistical relevance of the results, and a deep bioinformatic analysis that would require additional in vitro experiments to support the data obtained that is not the focus of this work.

8- In the xenograft experiments, Fig 6B and C would be better presented by showing each individual data point in addition to the mean and S.D.

As requested by the Reviewer, Figures 6B and 6C have been modified in the revised version of the manuscript (new revised Figure 7B-C).

9- the study will be further improved by investigating the effect of compound 22 administration in the prevention of MB formation in Math1-cre/Ptc1C/C mice. The experiment seems feasible since by 10 weeks of age, all animals develop MB (Z-J Yang et al. Cancer Cell 2008).

We agree with the Reviewer that the suggested experiment would improve our study, however the investigation of the effect of compound 22 administration, as well as of any potential novel drug, in the prevention of MB in vivo requires a deep and long study to set up the right treatment conditions in terms of amount of concentrations to be used, routes of administration and ability to cross the BBB. The efficacy of compound 22 in the prevention of MB development in Math-cre/Ptc1C/C mice will be one of the main goals in our future studies.

Minor corrections:

1- please revise the manuscript for use of English. For example, "This approach stands as a relevant to overcome the drug resistance... " for "This approach seems to effectively overcome drug resistance..."; "in a single and individual" for "into a single"; "antagonism at both Smo and Gli1" for "antagonism of Smo and Gli1";  "The Hedgehog signalling...." for "Hedgehog signalling or The Hedgehog pathway"; "Hh pathway" for "the Hh pathway". There are many more in accuracies that need to be fixed.

We thank the Reviewer for the advice and we modified the revised text as suggested. Further, the revised manuscript has been re-edited in several parts.

2- Please add a paragraph in the introduction describing the canonical Hh pathway for the reader's benefit.

We have now described the canonical Hh-pathway in the Introduction section.

3- Vismodegib resistance by activation of PKCiota should be included in the introduction.

We have now added this information.

Reviewer 2 Report

Hedgehog signaling plays diverse roles during the development of malignant diseases and small molecule inhibitors targeting the HH pathway member SMO have shown therapeutic efficacy in non-melanoma skin cancer and brain cancers. Rapid and frequent development of drug resistance is a severe problem for patients. The development of more efficient inhibitors is of high medical need to achieve better response rates and more durable therapeutic effects.

In this study, the authors pursue a strategy of simultaneously targeting SMO and oncogenic GLI transcription factors with the aim to effectively block oncogenic Hedgehog signaling and prevent or at least delay the development of drug resistance.

To this end, Severini and colleagues chemically modified the natural isoflavone scaffold to develop novel compounds with the ability to bind to and inhibit both the SMO and GLI oncogenes. This is a reasonable and innovative approach with relevance to medical cancer therapy.

The manuscript has been written concisely and the data are well presented.

However, several major points should be addressed before publication. In particular, the specificity of the new compounds as selective dual SMO/GLI inhibitors needs to be demonstrated in addition to the docking simulations presented.

1) Effective binding of e.g. compound 22 to SMO and GLI should be demonstrated, either in vitro or in cells expressing the two targets; 

2) The decrease in Hedgehog signal activity in response to compound treatment may also be caused by off-target cytotoxicity. Experiments such as rescue by e.g. GLI2 overexpression and for comparison, the use of non SHH medulloblastoma cells/cell lines should be included to reduce the concerns of off-target cytotox effects

3) Does compound 22 treatment interfere with GLI activity by preventing DNA binding or its nuclear localization? Answering these questions would strengthen the data and interpretation, particularly in terms of specificity and off-target effect.

4) Intratumoral injection of the inhibitor is problematic and should – if feasible - be replaced by systemic administration. What was the compound concentration injected into the tumor?

5) Smo-/- MEFs should not display pathway activity. Still the authors report activity of the compounds on GLI expression. This should be explained more clearly.

6) Does compound treatment interfere with ciliogenesis? The integrity of the primary cilium is critical for SMO-dependent Hedgehog signaling. Testing this would be an important control, as off-target effects disrupting ciliogenesis might unspecifically account for a reduction of Hedgehog signaling in response to treatment.

7) In addition to Gli1, it would be nice to also see other established Gli targets (e.g. Hhip and Ptch);

Author Response

Responses to Reviewers' comments:

Reviewer #2

Open Review

English language and style

( ) Extensive editing of English language and style required
( ) Moderate English changes required
(x) English language and style are fine/minor spell check required
( ) I don't feel qualified to judge about the English language and style

Yes

Can be improved

Must be improved

Not applicable

Does the introduction provide sufficient background and include all relevant references?

(x)

( )

( )

( )

Is the research design appropriate?

( )

(x)

( )

( )

Are the methods adequately described?

(x)

( )

( )

( )

Are the results clearly presented?

(x)

( )

( )

( )

Are the conclusions supported by the results?

( )

(x)

( )

( )

Comments and Suggestions for Authors

Hedgehog signaling plays diverse roles during the development of malignant diseases and small molecule inhibitors targeting the HH pathway member SMO have shown therapeutic efficacy in non-melanoma skin cancer and brain cancers. Rapid and frequent development of drug resistance is a severe problem for patients. The development of more efficient inhibitors is of high medical need to achieve better response rates and more durable therapeutic effects.

In this study, the authors pursue a strategy of simultaneously targeting SMO and oncogenic GLI transcription factors with the aim to effectively block oncogenic Hedgehog signaling and prevent or at least delay the development of drug resistance.

To this end, Severini and colleagues chemically modified the natural isoflavone scaffold to develop novel compounds with the ability to bind to and inhibit both the SMO and GLI oncogenes. This is a reasonable and innovative approach with relevance to medical cancer therapy.

The manuscript has been written concisely and the data are well presented.

We thank the Reviewer for appreciating the importance and soundness of our data.

However, several major points should be addressed before publication. In particular, the specificity of the new compounds as selective dual SMO/GLI inhibitors needs to be demonstrated in addition to the docking simulations presented.

1)         Effective binding of e.g. compound 22 to SMO and GLI should be demonstrated, either in vitro or in cells expressing the two targets; 

In order to address the point raised by Referee, we have tested the ability of compound 22 to bind SMO and GLI1. In the case of SMO, the direct binding was measured by titration with Bodipy-Cyclopamine (BC), a fluorescent derivative of the natural Hh inhibitor cyclopamine that competes with the same binding site in SMO as our newly designed isoflavones. BC displacement assay demonstrates that compound 22 binds SMO WT and its D473H mutant (which confers resistance following Vismodegib treatment) with similar affinity thus suggesting that compound 22 can overcome Vismodegib resistance (new Figure 4C).

In the case of Gli1, a similar binding assay has not been developed yet due to the lack of a reference ligand. However, to prove the potential ability of compound 22 to binds Gli1, we carried out a ChIP assay in Gli1-overexpressing MEF and we observed a significant reduction of the Gli1 recruitment into the promoter of Ptch1 following the treatment with compound 22 compared to control (new Figure 4E). It is worth noting that isoflavone 22 shares the scaffold as well as multiple pharmacophoric features with the reference GlaB, a well-known Gli1 antagonist for which we confirmed the direct binding to Gli1 by interaction with Lys350 and Lys340 in the Gli1 zinc-finger domain (Infante et al., EMBO J 2015). To confirm the results obtained by molecular docking of compound 22 to Gli1 zinc finger domain, we verified the effect of 22 on Gli1 mutated in Lys430 (Gli1 K340A), a residue involved in DNA binding and transcriptional function. As expected, different doses of isoflavone 22 induced a moderated reduction of Gli1-dependent transcriptional activation in HEK293T cells expressing ectopic Gli1 K340A and Gli1-dependent luciferase reporter compared to cells expressing Gli1 WT (new Figure 4D).

2)         The decrease in Hedgehog signal activity in response to compound treatment may also be caused by off-target cytotoxicity. Experiments such as rescue by e.g. GLI2 overexpression and for comparison, the use of non SHH medulloblastoma cells/cell lines should be included to reduce the concerns of off-target cytotox effects.

We thank the Reviewer for your suggestions. We tested the effect of compound 22 in several non HH-MB cell lines belonging to Group3 (D425, D458, HDMB03, D341) and Group3/4 (D283). As shown in Supplementary Figure S8, compound 22 does not impinge on the proliferation and cell death of the all tested cell models. Unfortunately, MB cells are not easily transfectable, and this issue makes it difficult to test rescue of Gli2, due also to high toxic effect caused by the combination of transfection and compound treatment.

3)         Does compound 22 treatment interfere with GLI activity by preventing DNA binding or its nuclear localization? Answering these questions would strengthen the data and interpretation, particularly in terms of specificity and off-target effect.

We thank the Reviewer for your suggestions. As described above, to verify the potential ability of compound 22 to binds Gli1, we performed a ChIP assay in Gli1-overexpressing MEFs and we observed a significant reduction of the Gli1 recruitment into the promoter of Ptch1 following the treatment with compound 22 compared to control (new Figure 4E). Further, we verified the effect of 22 on Gli1 mutated in Lys430 (Gli1 K340A) activity, a residue involved in DNA binding and transcriptional function of Gli1. As expected, different doses of isoflavone 22 induced a moderated reduction of Gli1-dependent transcriptional activation in HEK293T cells expressing ectopic Gli1 K340A and Gli1-dependent luciferase reporter compared to cells expressing Gli1 WT (new Figure 4D). Combining molecular modeling with in vitro assays we can substantiate our hypothesis that 22 binds Gli1.

Subcellular fractionation experiments indicate that compound 22 does not affect Gli1 subcellular localization (new Supplementary Figure S7D).

4)         Intratumoral injection of the inhibitor is problematic and should – if feasible - be replaced by systemic administration. What was the compound concentration injected into the tumor?

We agree with the Reviewer that systemic administration would improve our study. However, this experiment would require a deeper and long study to set up the right treatment conditions in terms of amount of concentrations to be used, routes of administration and ability to cross the BBB. The efficacy of compound 22 in inhibiting MB growth in vivo via systemic administration, as well as its efficacy in the prevention of MB development in HH-MB mouse model will be main goals in our future studies.

5)         Smo-/- MEFs should not display pathway activity. Still the authors report activity of the compounds on GLI expression. This should be explained more clearly.

We apologize for lack of clarity concerning this point. Although Gli1 mRNA in Smo-/- MEFs is low its levels are detectable, hence we decide to use this cell model to prove the inhibitory activity of isoflavones 21 and 22 on HH signaling independently of Smo. In this regard, we observed that our compounds, as well as GANT61 and ATO (two well known Gli inhibitors), but not Vismodegib (a Smo antagonist), reduce Gli1 mRNA levels in Smo-/- MEFs, thus suggesting the inhibitory activity of compounds 21 and 22 on Gli1 independently of Smo (new Supplementary Table 1 and new Supplementary Figure S4). This point is now discussed in the revised version of the manuscript.

6)         Does compound treatment interfere with ciliogenesis? The integrity of the primary cilium is critical for SMO-dependent Hedgehog signaling. Testing this would be an important control, as off-target effects disrupting ciliogenesis might unspecifically account for a reduction of Hedgehog signaling in response to treatment.

We thank the Reviewer for this suggestion. We have tested the possible effect of 22 on ciliogenesis, by analyzing both the number and the length of cilia in MEFs WT treated with the compound 22 at several concentrations. Immunofluorescence assay of the cilium marker IFT88 shows that shape, number and length of cilia are not affected by 22 (new Supplementary Figure S7), thus excluding an off-target effect on ciliogenesis.

7)         In addition to Gli1, it would be nice to also see other established Gli targets (e.g. Hhip and Ptch)

As requested by the Reviewer, expression analysis of other Gli-target genes, such as Ptch and Hhip, has been included in the revised version of the manuscript. Compound 22 significantly reduces the expression of Gli-dependent genes in all tested cell models (primary MB, Med1 and Daoy cells) (revised Figure 5C,D and new Figure 6C,G).

Reviewer 3 Report

A paper titled „A Smo/Gli multitarget Hedgehog pathway inhibitor impairs tumor growth“ by Severini et al is an interesting and well-writted paper which describes a novel Hedgehog pathway inhibitor. The newly synthetised inhibitor presented in this paper targets two proteins of the Hedgehog pathway simultaneously, increasing the efficacy of the compound by affecting both canonical and non-canonical Hedgehog signals. The authors show the effect of the compound on cell culture and in animal models, and on various models with impairment of Hedgehog signaling at different levels of the pathway.

There are only some minor comments, mostly typos, that I have noticed:

There is no reference to Figure2 A and B in the text. I guess it should be added somewhere around line 107.

Line 250 change the word „efficacy“ to „efficient“

Line 297 add the reference number

Line 316 correct the sentence, seems like some words are missing („...and used for short-term to keep...“)

Line 313: the cells were treated with DNase? Not trypsin? If DNase, why?

There is a small error in the legend of Figure S1, the box for 20 should be striped, not black.

On a general note, there is no information about the compounds' toxicity (on either cell lines or animal models). Did you test for toxicity? For example in a cell line independent of Hedgehog signaling?

Did you try using bioinfromatic tools to determine potential other targets of this compound? You have shown by molecular modelling that the compound should bind to both Smo and Gli proteins, but it is possible that the compound affects other proteins/pathways.

Author Response

Responses to Reviewers' comments:

Reviewer #3

Open Review

English language and style

( ) Extensive editing of English language and style required
( ) Moderate English changes required
(x) English language and style are fine/minor spell check required
( ) I don't feel qualified to judge about the English language and style

Yes

Can be improved

Must be improved

Not applicable

Does the introduction provide sufficient background and include all relevant references?

(x)

( )

( )

( )

Is the research design appropriate?

(x)

( )

( )

( )

Are the methods adequately described?

(x)

( )

( )

( )

Are the results clearly presented?

(x)

( )

( )

( )

Are the conclusions supported by the results?

(x)

( )

( )

( )

Comments and Suggestions for Authors

A paper titled „A Smo/Gli multitarget Hedgehog pathway inhibitor impairs tumor growth“ by Severini et al is an interesting and well-writted paper which describes a novel Hedgehog pathway inhibitor. The newly synthetised inhibitor presented in this paper targets two proteins of the Hedgehog pathway simultaneously, increasing the efficacy of the compound by affecting both canonical and non-canonical Hedgehog signals. The authors show the effect of the compound on cell culture and in animal models, and on various models with impairment of Hedgehog signaling at different levels of the pathway.

We thank the Reviewer for appreciating the importance and soundness of our data.

There are only some minor comments, mostly typos, that I have noticed:

There is no reference to Figure2 A and B in the text. I guess it should be added somewhere around line 107.

Figure 2A and 2B are indicated in the revised version of the manuscript.

Line 250 change the word „efficacy“ to „efficient“

We modified the revised text as suggested.

Line 297 add the reference number

The reference number has been added in the revised version of the manuscript.

Line 316 correct the sentence, seems like some words are missing („...and used for short-term to keep...“)

We corrected the sentence as suggested.

Line 313: the cells were treated with DNase? Not trypsin? If DNase, why?

In primary culture of tumor cells, preparing a single cell suspension is critical for successful cell isolations. Cells may sometimes appear "clumpy" when they have been exposed to enzymatic tissue dissociation (i.e trypsin, that stresses MB primary cells if used during the disgregation phase). Moreover, the cell clumps can accelerate the rate of cell death within the cell culture, resulting in the release of DNA molecules from the dying cells that can clump neighboring cells together.  Adding the DNase I into the cell culture can minimize the presence of free-floating DNA fragments and cell clumps, without toxic effects.

There is a small error in the legend of Figure S1, the box for 20 should be striped, not black.

We corrected the legend of Figure S1

On a general note, there is no information about the compounds' toxicity (on either cell lines or animal models). Did you test for toxicity? For example in a cell line independent of Hedgehog signaling?

We thank the Reviewer for the suggestions. We tested the effect of compound 22 in several non HH-MB cell lines belonging to Group3 (D425, D458, HDMB03, D341) and Group3/4 (D283). As shown in Supplementary Figure S8, compound 22 does not impinge on the proliferation and cell death of the all tested cell models.

Did you try using bioinfomatic tools to determine potential other targets of this compound? You have shown by molecular modelling that the compound should bind to both Smo and Gli proteins, but it is possible that the compound affects other proteins/pathways.

Although we cannot exclude the activity of compound 22 on additional target, we demonstrated that this compound does not affect luciferase activity driven by Hh-unrelated (i.e. Jun/AP1) and Hh-related (i.e. Wnt/β-catenin) pathway (new Supplementary Figure 5), and does not affect protein levels of others Hh regulators (i.e. HDAC1, ERAP1, Itch, β-catenin) (new Figure 5E), further indicating its selectivity for Hh/Gli signaling.

RNASeq, as suggested by the reviewer, could definitely give many information about the possible effect of compound 22 on other signaling pathways. Nevertheless, this would require many replicates, especially for tumor primary cell culture, to strengthen the statistical relevance of the results, and a deep bioinformatic analysis that would require additional in vitro experiments to support the data obtained that is not the focus of this work.

Round 2

Reviewer 1 Report

The revised manuscript by Severini et al. provides a strong case for Hedgehog pathway specificity of the novel isoflavone-based compounds 21 and 22, both in vitro in normal and cancer cells as well as in animal models of medulloblastoma. In my opinion, this is now acceptable for publication.

I would like to ask the authors to spell check the manuscript once more (ex. NIH 3t£ in methods).

Author Response

Comments and Suggestions for Authors

The revised manuscript by Severini et al. provides a strong case for Hedgehog pathway specificity of the novel isoflavone-based compounds 21 and 22, both in vitro in normal and cancer cells as well as in animal models of medulloblastoma. In my opinion, this is now acceptable for publication.

I would like to ask the authors to spell check the manuscript once more (ex. NIH 3t£ in methods).

We thank the Reviewer for appreciating the importance and soundness of our work. The revised manuscript has been re-edited, as suggested by Reviewer.

Reviewer 2 Report

The reviewer appreciates the detailed point-by-point response by the authors. The revised version has been strengthened significantly and the major points addressed. As minor point, the authors should point out the limitation of the intra-tumoral injection treatment in the discussion (lines 365-371).

Author Response

Comments and Suggestions for Authors

The reviewer appreciates the detailed point-by-point response by the authors. The revised version has been strengthened significantly and the major points addressed. As minor point, the authors should point out the limitation of the intra-tumoral injection treatment in the discussion (lines 365-371).

We thank the Reviewer for the positive comments on our revised manuscript. We have pointed out the limitations of intratumoral injection, as suggested by Reviewer.
